# The Impact of Ca^2+^ on Intracellular Distribution of Hemoglobin in Human Erythrocytes

**DOI:** 10.3390/cells12182280

**Published:** 2023-09-15

**Authors:** Leonid Livshits, Sari Peretz, Anna Bogdanova, Hiba Zoabi, Harel Eitam, Gregory Barshtein, Cindy Galindo, Yuri Feldman, Ivana Pajić-Lijaković, Ariel Koren, Max Gassmann, Carina Levin

**Affiliations:** 1Red Blood Cell Research Group, Vetsuisse Faculty, Institute of Veterinary Physiology, University of Zurich, 8057 Zürich, Switzerland; annab@access.uzh.ch (A.B.); maxg@access.uzh.ch (M.G.); 2Pediatric Hematology Unit, Emek Medical Center, Afula 1834111, Israel; sari_pe@clalit.org.il (S.P.); koren_a@clalit.org.il (A.K.); levin_c@clalit.org.il (C.L.); 3Laboratory Division Unit, Emek Medical Center, Afula 1834111, Israel; hiba.zoabi@gmail.com (H.Z.); eitam_ha@clalit.org.il (H.E.); 4The Bruce and Ruth Rapaport Faculty of Medicine, Technion–Israel Institute of Technology, Haifa 3200003, Israel; 5The Zurich Center for Integrative Human Physiology (ZIHP), 8057 Zürich, Switzerland; 6Biochemistry Department, The Faculty of Medicine, The Hebrew University of Jerusalem, Jerusalem 9112102, Israel; gregoryba@ekmd.huji.ac.il; 7Institute of Applied Physics, The Hebrew University of Jerusalem, Jerusalem 9190401, Israel; cindy.galindohur@mail.huji.ac.il (C.G.); yurif@mail.huji.ac.il (Y.F.); 8Department of Chemical Engineering, University of Belgrade, 11000 Beograd, Serbia; iva@tmf.bg.ac.rs

**Keywords:** hemoglobin distribution, red blood cells, hemoglobin A2, calcium

## Abstract

The membrane-bound hemoglobin (Hb) fraction impacts red blood cell (RBC) rheology and metabolism. Therefore, Hb–RBC membrane interactions are precisely controlled. For instance, the signaling function of membrane-bound deoxy-Hb and the structure of the docking sites in the cytosolic domain of the anion exchanger 1 (AE-1) protein are well documented; however, much less is known about the interaction of Hb variants with the erythrocyte’s membrane. Here, we identified factors other than O_2_ availability that control Hb abundance in the membrane-bound fraction and the possible variant-specific binding selectivity of Hb to the membrane. We show that depletion of extracellular Ca^2+^ by chelators, or its omission from the extracellular medium, leads to membrane-bound Hb release into the cytosol. The removal of extracellular Ca^2+^ further triggers the redistribution of HbA0 and HbA2 variants between the membrane and the cytosol in favor of membrane-bound HbA2. Both effects are reversible and are no longer observed upon reintroduction of Ca^2+^ into the extracellular medium. Fluctuations of cytosolic Ca^2+^ also impact the pre-membrane Hb pool, resulting in the massive transfer of Hb to the cellular cytosol. We hypothesize that AE-1 is the specific membrane target and discuss the physiological outcomes and possible clinical implications of the Ca^2+^ regulation of the intracellular Hb distribution.

## 1. Introduction

Hemoglobin (Hb), by far the most abundant protein in red blood cells (RBCs), is known primarily for its key function in blood gas exchange and transport. However, its cellular activities are not limited to this function. Pre-membrane Hb and cytosolic Hb pools have distinct roles in controlling the metabolic activity of glycolytic enzymes and nitric oxide (NO) production and in the rheological properties of RBCs [1,2,3,4]. Interactions of deoxy-Hb with the cytosolic domain of transmembrane band 3 protein controls the activity of glycolytic enzymes [3,5]; deoxy-Hb catalyzes nitrite conversion to NO [6]. Mean cell Hb concentration, its aggregation—as in the case of pathological variants such as HbS and HbC—and the thickness of the pre-membrane Hb layer affect membrane stability and RBC rheology [7].

In the RBCs of healthy donors, 0.5–12% of the Hb forms a membrane-bound pool, while the rest of the Hb remains in the cytosol [2]. The considerable variation in the extent of the membrane-bound Hb fraction depends on a plethora of (patho)physiological conditions, such as Hb O_2_ saturation, oxidative stress, cellular Hb concentration, charge, stability of Hb variants, and elevated Hb oxidation to methemoglobin, which are all reported to affect Hb binding to the membrane [8,9,10,11,12,13]. The different methodologies used to isolate and detect RBC membranes may also contribute to the variation (summarized in Ref. [9]). 

Several types of Hb–membrane interactions are known. These include the following: (a) electrostatic binding of deoxy-Hb to the cytoplasmic domain of band 3 anion transport protein (anion exchanger (AE-1) protein) [14,15]; (b) covalent crosslinking with the membrane components via disulfide bonds, and (c) adsorption to membrane lipids via hydrophobic interactions [16,17].

To the best of our knowledge, no previous attempts have been made to determine whether the distribution of Hb isoforms/variants between the membrane and the cytosol in healthy individuals is random or if preference is given to specific variants over others. In general, Hb in the RBCs of healthy adults is present in three main isoforms [18]. HbA0 makes up >96% of the Hb, whereas HbA2 typically does not exceed 2–3.5%, complemented by the even less abundant remnants (~1%) of fetal Hb (HbF), which is the dominant Hb isoform in the fetus. In contrast to HbF, HbA0 and HbA2 are ubiquitously distributed in most of the circulating cells [19]. The Hb isoforms are not located uniformly within the RBCs; due to its positive charge, HbA2 has a higher affinity for RBC membrane proteins than the other Hb isoforms [20]. The potential regulatory or catalytic role of the HbA2 variant in RBC membranes has never been explored. It is suggested that HbA2 may be involved in controlling RBC morphology by regulating the activity of the K–Cl co-transport system or by tuning cell pH [21]. In addition to its involvement in oxygen transport, HbA2′s signaling activity has been speculated but not confirmed [19,21]. Furthermore, if HbA2 is a sensor that responds to stimulation via redistribution between the membrane-bound and cytosolic pools, it is not clear what these stimuli are.

In the present study, we examined the distribution of HbA0 and A2 variants between the cytosolic- and membrane-bound pools, focusing on the possible variant selectivity in these responses to plasma-borne stressors. We identified the distinct roles of extracellular and intracellular Ca^2+^ as stimulants and monitored the potential outcome of Ca^2+^-induced redistribution of Hb between the membrane-bound and cytosolic pools on the RBC’s morphology and physiological parameters, such as membrane stability and metabolic readouts.

## 2. Materials and Methods

### 2.1. Blood Samples

The remains of adults’ fresh blood samples sent for routine analysis to the central laboratory at Emek Medical Center in K_3_EDTA-, citrate- or heparin sulfate-supplemented tubes in the years 2021–2023 were chosen at random. The study was performed in accordance with the Declaration of Helsinki and approved by the Emek Medical Center ethics committee (EMC-0085-21).

### 2.2. Buffers and Chemicals

Commercial phosphate-buffered saline, either Ca^2+^/Mg^2+^-free (PBS) or with 0.9 mM Ca^2+^ and 0.5 mM Mg^2+^ (DPBS), was purchased from Biological Industries (Haemek, Israel) and Sartorius (Göttingen, Germany), respectively. Plasma-mimicking buffer (PMB) contained the following: 140 mM of NaCl, 4 mM of KCl, 0.75 mM of MgSO_4_, 10 mM of glucose, 0.015 mM of ZnCl_2_, 0.2 mM of glycine, 0.2 mM of sodium glutamate, 0.2 mM of alanine, 0.1 mM of arginine, 0.6 mM of glutamine, and 20 mM of HEPES, adjusted to pH 7.4 with imidazole and then supplemented with 0.01% bovine serum albumin (BSA). When required, 2 mM of CaCl_2_ or 5 mM of ethylenediaminetetraacetic acid (EDTA) was added. Citrate phosphate dextrose adenine solution (CPDA-1) was purchased from Macopharma (Tourcoing, France). All other chemicals were bought from Sigma-Aldrich Israel (Rehovot, Israel). 

### 2.3. Hb Variant Analysis

Hb variants were detected by HPLC. The VARIANT™ II β-thalassemia Short Program (Bio-Rad, Hercules, CA, USA) method, which separates Hb variants by cation-exchange chromatography using a salt gradient, was used, with the calibrators and controls provided by the manufacturer with every batch. The analysis consisted of monitoring retention times, area percentages, and concentrations of various peaks and windows for different Hb variants: HbF (retention time of 1.1 min with 0.98–1.2 min window), HbA0 (2.5 min and 2.0–3.0 min), HbA2 (3.65 min and 3.57–3.75 min), and minor peaks, such as P2 (1.39 min and 1.28–1.5 min) and P3 (1.7 min and 1.5–1.9 min). 

### 2.4. RBC Membrane Preparation

To isolate the membrane fraction of RBCs, we adapted the protocol reported by Ghashghaeinia et al. [22]. Briefly, a 150 µL aliquot of the blood sample was incubated with 20 volumes of ice-cold HEPES-based hypoosmotic solution (20 mM of HEPES/NaOH, 1 mM of PMSF, pH 7.4) for 10 min and then centrifuged (4 °C, 14,000× *g*, 20 min). This procedure was repeated three times prior to the measurements of Hb content or Hb isoform distribution. Hb concentrations in intact RBCs and RBC membranes were measured using an MRC Spectro V-18 spectrophotometer (absorbance at 575 nm) and then evaluated in accordance with a prior calibration with known concentrations of Hb. After the RBCs or the membranes were isolated and washed, 5 μL of intact RBCs or 30 μL of RBC membranes were diluted in 4 mL DDW and measured using the spectrophotometer. Hb isoform distributions were examined as described in Section 2.3.

### 2.5. Morphological Characterization Using Cell Flow-Properties Analyzer (CFA)

To examine morphological changes, we used a computerized CFA [23,24,25]. Briefly, 50 μL of RBC suspension (1% hematocrit, in the same medium used for pretreatment) was placed into a flow chamber with an uncoated glass slide. The RBCs were allowed to adhere to the slide surface for 10 min before capturing images. We captured at least 10 fields (more than 1150 cells in total) for each sample. The CFA image analysis program can automatically measure the major and minor cellular axes for individual cells. A major-to-minor axis ratio of 1 reflects a round RBC. Cells with a ratio above 1.25 were removed from further analysis. To compare RBC shapes in different examinations, we evaluated the projected area of each cell by multiplying the major and minor axis values.

### 2.6. Glucose Consumption, Lactate Release, and K^+^ Leakage Studies

After removal of the plasma and the buffy coat, the cells were incubated in PMB with or without 2 mM of Ca^2+^, or with both 2 mM of Ca^2+^ and 5 mM of EDTA (hematocrit ~20%) for 2 h. Then, the cells were centrifuged at 1700× *g* for 5 min, the supernatant was discarded, and the cells were resuspended in a fresh medium. The cells were quickly mixed, and basal levels of extracellular K^+^, glucose, and lactate were detected using a GEM^®^ Premier™ 5000 blood gas analyzer (Werfen, Bedford, MA, USA). The cells were then incubated for 4 h at 37 °C in a shaker, and the measurements of extracellular K^+^, glucose, and lactate levels in PMB were repeated. Total Hb levels in the RBC suspension were also assessed prior to the one-point test with the blood gas analyzer. Changes in K^+^, glucose, and lactate concentrations in PMB representing glucose conversion to lactate, and K^+^ loss from RBCs over 4 h were then expressed in mmole per gram, Hb per hour. 

### 2.7. Labeling Experiments

Morphological changes, membrane potential, and intracellular Ca^2+^ levels were investigated via flow cytometry. The intracellular Ca^2+^ dye Fluo-4 AM (1 mM stock, Thermo Fisher Scientific, Waltham, MA, USA) and the voltage-sensitive dye bis (1,3-dibutylbarbituric acid) trimethine oxonol (DiBAC4(3); 0.2 mM stock, Molecular Probes, Eugene, OR, USA) were used. Briefly, RBCs were washed free of the plasma and buffy coat, and 1 µL of packed cells was resuspended in 1 mL of the desired buffer and incubated for 1 h at 37 °C. Then, 1 µL of one of the dyes was added to 1 mL of the sample, and the cell suspensions were incubated for another hour at 37 °C in the dark. Thereafter, fluorescence intensity was measured in stained RBCs using a Navios EX flow cytometer (Beckman Coulter, Brea, CA, USA). Measurements were repeated at least twice (>30,000 measured cells) and averaged for each condition. All data were analyzed using Kaluza Analysis Software (Beckman Coulter, https://www.beckman.co.il/flow-cytometry/software/kaluza, Version 2.1.00001.20653, built on 8 March 2018).

### 2.8. Separation on a Percoll Density Gradient

Fractionation on a self-formed isotonic continuous Percoll density gradient was performed as described [26]. Briefly, 1 mL of whole blood was gently layered on top of 13 mL of a 90% isotonic Percoll mixture (consisting of nine parts of commercial Percoll (GE HealthCare, Chicago, IL, USA) and one part of 10× concentrated PMB) supplemented with a final 0.1% BSA and 2 mM of CaCl_2_. RBCs were separated out via centrifugation at 18,514× *g* for 60 min (minimal acceleration/deceleration) at 30 °C (Eppendorf Centrifuge 5810R, F-34-6-38 rotor supplemented with specific adapters for 15 mL Falcon tubes).

### 2.9. Statistics

Data for the entire study were analyzed using GraphPad 5 software. The normality of distribution of the values obtained in each experimental set was evaluated using the Shapiro–Wilk test, and those with *p* > 0.05 were considered normally distributed. For those parameters showing normal distribution, paired-matched values were compared using a paired Student’s *t*-test. For the datasets that were not normally distributed, the Wilcoxon signed-rank test was used. For all analyses, a two-tailed test with *p* < 0.05 was accepted as statistically significant. For more details, see figure legends.

## 3. Results

### 3.1. Effect of Extracellular Constituents on Hb Distribution in the Membrane

Figure 1 presents the membrane distribution of Hb isoforms in RBCs exposed to routinely used anticoagulants and storage solution CPDA-1. In view of the minimal cell fraction of HbF and its mostly homogeneous intracellular distribution (Table 1), we concentrated on the HbA2-to-HbA0 isoform ratios in intact RBCs (which predominantly correlate with the distribution of these Hb isoforms in the RBC cytosol) and in their membrane compartment. Figure 1A,B present the HbA2/HbA0 ratios in RBCs collected in K_3_EDTA- and heparin-supplemented tubes, respectively. Intriguingly, the HbA2 fraction in the pre-membrane pool was significantly higher than those in intact RBCs and in the cytosolic compartment. Furthermore, HbA2 abundance in the membrane was strongly dependent on the type of supplemented anticoagulant (Figure 1C), with maximal values found in RBCs collected into K_3_EDTA-supplemented tubes. Moreover, as demonstrated in Figure 1D, RBC maintenance in the routine storage solution CPDA-1 led to an increase in the membrane fraction of HbA2 compared to that in the cytosol or intact cells. The pre-membrane HbA2/HbA0 ratio remained constant during prolonged storage of RBC concentrates (i.e., for 3, 14, and 28 days at 4 °C). Intriguingly, 28 days of storage at 4 °C in CPDA-1 medium caused a decrease in the total HbA2 fraction in intact RBCs. This finding is in partial agreement with the findings of Hildrum et al. [27]; however, the mechanism underlying this alteration is not clear and needs to be further investigated.

In the next set of experiments, we examined the size and content of the pre-membrane Hb pool in RBCs from EDTA-preserved whole blood after the plasma had been substituted with one of the routinely used isotonic buffers, DPBS or PMB (their chemical compositions are detailed in Table 2). In a separate set of experiments, we assessed the possible temperature dependence of the HbA2/A0 distribution between the membrane and the cytosolic pool. To do so, washed RBCs were resuspended in a Ca^2+^-free EDTA-containing medium and incubated for two hours at either of the following temperatures: 4, 25, 37, or 42 °C. No effect of temperature on the distribution of the Hb variant between the membrane and the cytosol could be confirmed (Appendix A). Based on these data, we have chosen 37 °C as the optimal temperature for further investigation of the underlying mechanism. The time dependence of the HbA2/HbA0 redistribution in the pre-membrane pool was then examined. We observed at least a two-phase process of redistribution of the HbA0 and HbA2 variants at the membrane (Figure 2A). First, a sharp decrease of the pre-membrane HbA2/HbA0 was detected after 15 min incubation in either DPBS or PMB. Then, the membrane fraction of HbA2 in the bulk pre-membrane Hb pool continued to gradually decrease to a minimum at 2 h. There were no significant variations for the membrane Hb ratios in DPBS- vs. PMB-treated RBCs, except after 1 h of maintenance. Longer incubations, for 6–24 h, were associated with a modest partial recovery of the HbA2/HbA0 ratio in the pre-membrane fraction. Slight increases were observed at 6 and 24 h. Based on these findings, 2 h was chosen as the optimal time point for further investigation of the short-term mechanism. Numerically, the membrane-bound HbA2/HbA0 fractions in samples maintained with DPBS or PMB were twice as small than in those suspended in the EDTA-containing plasma after 2 h of incubation (Figure 2B). Intriguingly, the observed alteration in the pre-membrane HbA2 fraction was associated with a significant rise in the concentration of total Hb in the pre-membrane Hb pool in both DPBS and PMB (Figure 2C).

### 3.2. The Key Role of Extracellular Ca^2+^ in HbA2 Enrichment of the Membrane-Bound Hb Pool 

Analysis of the experimental conditions supporting the enrichment of the pre-membrane Hb pool with HbA2 summarized in Table 2 and Table 3 suggested that extracellular divalent cations play an important role in it, while plasma proteins are not involved in the preferential HbA2 binding at the membrane. To verify this assumption, the cells were incubated in Ca^2+^/Mg^2+^-free PBS with specific supplementation of the chloride salts of either Fe^3+^ (0.1 mM), Zn^2+^ (0.02 mM), Ca^2+^ (2 mM), Cu^2+^ (1 mM), or Mg^2+^ (1 mM) at near-physiological concentrations (Figure 3). The presence of Ca^2+^, but not of the other cations tested, resulted in the observed significant reduction in the HbA2/HbA0 ratio in the membrane-bound pool. Moreover, the selective preferential recruitment of HbA2 to the membrane showed dose dependence with the extracellular Ca^2+^ concentrations within the 0–2 mM range in both PBS and PMB (Figure 4).

### 3.3. Possible Interrelated Effect of Hypocalcemia on Hb Distribution and RBC Structural and Metabolic Features 

The following sets of experiments were designed to explore (a) whether the chelation of extracellular Ca^2+^ triggers Hb pre-membrane redistribution and (b) if the recovery of physiological concentrations of Ca^2+^ will rescue the native HbA2 distribution. For this set of experiments, we collected RBCs into heparin-supplemented tubes, thus keeping the native extracellular Ca^2+^ concentration unchanged. EDTA (final concentration of 5 mM) was then added to the heparinized whole blood samples (Figure 5, upper panel). We found that chelation of extracellular Ca^2+^ leads to a significant increase in the membrane HbA2 fraction. Moreover, when Ca^2+^ was supplemented after EDTA removal (by the cycles of washes with heparinized plasma) and the cells were incubated for an additional 2 h in the presence of extracellular Ca^2+^, we found a complete recovery of HbA2 distribution to that in the initial (pre-EDTA supplemented) plasma. Total Hb content in the membranes mirrored these changes: Ca^2+^ isolation decreased the abundance of Hb in the membrane-bound pool, and the replenishing of Ca^2+^ led to the recovery of the membrane-bound Hb fraction (Figure 5, bottom panel). These observations were supported by experiments performed using PBS or PMB as an exterior milieu instead of blood plasma. Thus, no specific blood plasma constituents (except Ca^2+^) are required to regulate the abundance of Hb in the pre-membrane pool or its variant-specific distribution. 

Because of the previously reported contribution of membrane-associated Hb to cellular stability and metabolic processes, we further tested whether the 2 h of hypocalcemia (by incubating the cells in a Ca^2+^-free medium or in the presence of EDTA) and the corresponding membrane Hb redistribution would have an effect on these parameters. Examination of CFA images indicated a barely noticeable impact on the shape and morphology of RBCs (Figure 6A,B). In addition, a slight effect on RBC hydration (as assessed through separation on a Percoll density gradient) was found. In parallel, we revealed a significant increase in membrane permeability (estimated with elevated K^+^ leakage) under Ca^2+^-free or chelated conditions and complete membrane recovery with Ca^2+^ reconstitution (Table 4).

No specific effect of the absence of Ca^2+^ on metabolic properties (glucose consumption or lactate release) was observed in the Ca^2+^-added vs. lacking RBCs (Table 4). In contrast, we observed significant inhibition of both processes with EDTA-mediated chelation and their recovery with Ca^2+^ replenishment. The observed differences in Ca^2+^-free vs. chelating media may reflect the requirement for additional tri-/bivalent cations for the physiological regulation of metabolic processes. 

### 3.4. Possible Intracellular Mechanism(s) Governing Ca^2+^-Mediated Hb Translocation 

We then set out to clarify the possible intracellular mechanism(s) underlying the observed phenomenon of Ca^2+^-mediated Hb translocation. First, we tested whether the observed Hb redistribution is activated by Ca^2+^ uptake into the cells. We measured the size and isoform composition of the pre-membrane Hb pool after RBC incubation with increasing concentrations of Ca^2+^ in the presence of the Ca^2+^ ionophore A23187 (at 10 µM final concentration) (Figure 7). The uptake of Ca^2+^ in the cells was confirmed by the simultaneous monitoring of intracellular Ca^2+^ change via flow cytometry using Fluo-4 fluorescence (Table 5). We found a significantly higher fraction of HbA2 in the RBC membrane Hb pool when the cells were incubated in the presence of 0.125 or 0.5 mM extracellular Ca^2+^ along with A23187. The less pronounced action of the higher (2 mM) Ca^2+^ concentration in the medium could be explained by the shedding of A23187 due to vesiculation caused by acute Ca^2+^ overload. 

We have tested if Ca^2+^ supplementation results in a change in the transmembrane potential which, in turn, could trigger Hb trafficking to the membrane to balance the membrane charge. We took two approaches to test this hypothesis: (a) using the specific voltage-sensitive fluorescent dye DiBAC4(3) [28,29] and (b) by varying KCl concentrations in the extracellular buffer in the presence of the K^+^ ionophore valinomycin [30]. Specifically, DiBAC4(3) enters depolarized cells with the corresponding binding of intracellular proteins and elevation in fluorescence [31]. Using the first approach, we found small but significant Ca^2+^-dose-dependent changes in the transmembrane potential (Figure 8A). Using the second approach, the transmembrane potential was affected by maintaining the RBCs in media with increasing concentrations of KCl (Figure 8B). KCl substituted the equivalent fractions of NaCl in the presence of valinomycin (1 µM final concentration). In accordance with the Nernst equation, the transmembrane potential in the exposed RBCs increases with gradually increasing concentrations of KCl to a maximum of + 10 mV in 150 mM of KCl medium [29]. Intriguingly, we observed reduced and minimally varying membrane HbA2/HbA0 values in all spectra of the examined NaCl/KCl concentrations, raising doubts as to the role of transmembrane potential in regulating Hb membrane distribution. 

### 3.5. AE-1 as a Membrane Target for Ca^2+^-Induced Hb Binding

Previous studies have reported that AE-1 is a primary target membrane molecule for Hb binding to the membrane [8,14,15]; we tested whether alterations in this exchanger’s structure or activity would affect Hb distribution. We treated the cells with a well-known AE-1 inhibitor, DIDS (50 µM, 30 min in 37 °C), or with 2 mM of ZnCl_2_ (1 h, 25 °C), which, at this supraphysiological concentration, causes mild AE-1 clustering [32]. Intriguingly, we found that AE-1 clustering in the presence of Ca^2+^ returned the HbA2 membrane fraction to the corresponding values of calcium-unexposed RBCs (Figure 9). In contrast, inhibition of AE-1-mediated transport with DIDS minimally affected the Hb variant ratio. These results suggest the possible involvement of AE-1 in the mechanism of calcium-mediated Hb trafficking.

## 4. Discussion

We examined the regulation of the intracellular distribution of Hb between the membrane and the cytosol, namely, the size of the pre-membrane Hb pool and its isoform composition. Our main finding was the unique and distinct roles of intracellular and extracellular Ca^2+^ in the control of pre-membrane Hb pool size and composition (Figure 10). Reversibility of the effects indicated that the pre-membrane Hb pool in the RBCs of healthy donors is largely populated by non-covalently-bound Hb, both A0 and A2 isoforms, and not with covalently-bound Hb, as reported previously (see, for instance, Ref. [8]). Redistribution of HbA2 and HbA0 variants in response to chelation of extracellular Ca^2+^ was a biphasic process, involving Hb release from the membrane through acute (minute range)–intensive and then prolonged–gradual stages. Extracellular Ca^2+^ supported a higher level of Hb in the pre-membrane pool with lower HbA2 abundance, and an increase in cytosolic Ca^2+^ induced Hb translocation to the cytosolic compartment. Selectivity for HbA2 control in the pre-membrane pool may be associated with its higher binding affinity [20].

We also assumed a correlative effect of changes in the extracellular Ca^2+^ level on the RBCs’ physiological changes. We found that Ca^2+^ removal, with the corresponding Hb rearrangement, specifically affects RBC membrane stability and cellular metabolism. These findings may be crucial for understanding the molecular mechanisms controlling Ca^2+^-mediated Hb presence in the membrane, mainly because of the possible involvement of membrane proteins in building the cytoskeleton or directly regulating its stability (discussed in Section 4.1). However, a deeper exploration of the mechanisms governing the involvement of the membrane Hb in physiological processes was outside the scope of the current work and warrants future research. 

### 4.1. Possible Membrane Targets for Ca^2+^-Regulated Presence of Hb 

The most intriguing question that arises from our data is how Ca^2+^ regulates Hb’s presence in the membrane. Our experiments with A23187 showed that Ca^2+^ directly decreases the protein’s translocation to the membrane. Whether this process is controlled by the direct interaction of Hb and Ca^2+^ (as proposed in [33,34]), or if there is some essential involvement of other cytosolic molecules, is still an open question, warranting further investigation. However, the increase in Hb with elevated Ca^2+^ implies that the process is initiated via stimulation of an extracellular target. Band 3 protein (AE-1) may be viewed as an ideal candidate, supported by evidence from the literature as well as from the current study. First, AE-1 is a known target for Hb–membrane binding [8,14,15]. For example, competition between Hb and metabolic enzymes for attachment to AE-1 is one of the central regulators of glucose metabolism in RBC [3,5]. Second, HbA2 has a more positive charge than the other Hb variants, providing increased affinity for membrane AE-1 [19]. Finally, AE-1 itself is a Ca^2+^-binding protein [35,36,37]. Taken together, we cannot exclude the role of AE-1 in these processes, but further studies are needed to confirm the hypothesis.

There may be other membrane components regulated by intracellular Ca^2+^ that can impact the presence of Hb on the membrane. Hb has been reported to interact with spectrin [38], mainly as part of a high-molecular-weight cytoskeletal complex that also contains ankyrin and band 4.2 [8]. These membrane proteins may be direct targets of Ca^2+^, or at least affected by Ca^2+^-activated proteins [39,40,41]. Ca^2+^ activation of the calmodulin 4.1R has been found to reduce the latter’s affinity for its cytoskeletal targets, thus affecting the spectrin/actin network and cell shape [42]. Moreover, the possible involvement of intracellular rearrangement of Na^+^ and K^+^ in Hb redistribution due to functional alteration of the Na^+^/K^+^ pump should also be considered [43]. The specific interactions of Hb and Ca^2+^ with anionic phosphatidylserine [44,45] may indicate that the more positive HbA2 variant shares binding sites for Ca^2+^ in stabilizing the membrane’s phospholipid distribution. Therefore, the identification of which Ca^2+^-initiated molecular cascades are primarily involved in Hb reorganization in the membrane is one of our paramount tasks. 

### 4.2. Suggestions for Laboratory and Clinical Practice 

The current results may have significant clinical importance. One possible consequence might be to reconsider the strategy for choosing specific chelators for routine blood collection, the maintenance of RBCs prior to laboratory examination, and better interpretation of the clinical analytical data. Specifically, EDTA is a frequently used metal chelator for blood sample anticoagulation [46]. Due to its higher Ca^2+^ ion binding constant, EDTA could be used at lower concentrations than, for example, citric acid [47] (at the same time explaining the limited effect of citrate on Hb distribution (Figure 1C)). However, EDTA chelating activity is not limited to the buffer’s cations; although EDTA does not cross the intact membrane of erythrocytes, it may bind to the external membrane surface and remove ~90% of the membrane-bound Ca^2+^ [48]. As a result, some cellular features initiated by Ca^2+^-binding to the RBC surface or its uptake are immediately altered with EDTA supplementation. For example, the appearance of hundreds of thin extrusions on the RBC membrane was observed by Pinteric et al. [49] after EDTA supplementation to an RBC sample. It is important to note that these membrane abnormalities were completely recovered by Ca^2+^ supplementation, meaning that these RBC modifications are due to the specific removal of Ca^2+^ and not to the Ca^2+^-independent side effects of the chelator. This phenomenon may provide a partial explanation for the observed increase in membrane permeability observed in the current study (Table 4).

The proposed mechanism is essential for understanding and preventing numerous post-transfusion events. Routinely used storage solutions are Ca^2+^-free; therefore, massive transfusion of Ca^2+^-free RBC suspension (especially in severe cases) can cause an acute decrease in the normal level of Ca^2+^ in patients. Because of the dose–response effect of Ca^2+^ on Hb cellular distribution, the subsequent outcome on Hb-associated RBC features may appear, even with relatively small fluctuations of plasma Ca^2+^ in recipients immediately after blood transfusions. Several post-transfusion vascular complications have been reported [50], and the possible influence of the proposed mechanisms on the pathogenesis of these events should therefore be further investigated. This is also relevant for the advanced attempts to incorporate EDTA-based chelating therapies (for example, the Trial to Assess Chelation Therapy (TACT) projects) that require detailed information on the possible changes in RBCs to prevent side-effect complications [51].

## 5. Conclusions

Novel mechanisms for the Ca^2+^ regulation of Hb localization in RBCs are presented. We show that the total content and variant distribution of Hb are strongly associated with the presence of Ca^2+^ in the extracellular milieu, and any fluctuation in Ca^2+^ level may significantly affect the membrane fraction of Hb. Fluctuations of cytosolic Ca^2+^ also impact the pre-membrane Hb pool resulting in massive transfer of Hb to the cellular cytosol. We also suggest an interrelationship between the revealed mechanisms and Ca^2+^-mediated changes in RBC structural and metabolic properties. A detailed delineation of the causes and consequences of this interaction awaits further clarification.

## Figures and Tables

**Figure 1 cells-12-02280-f001:**
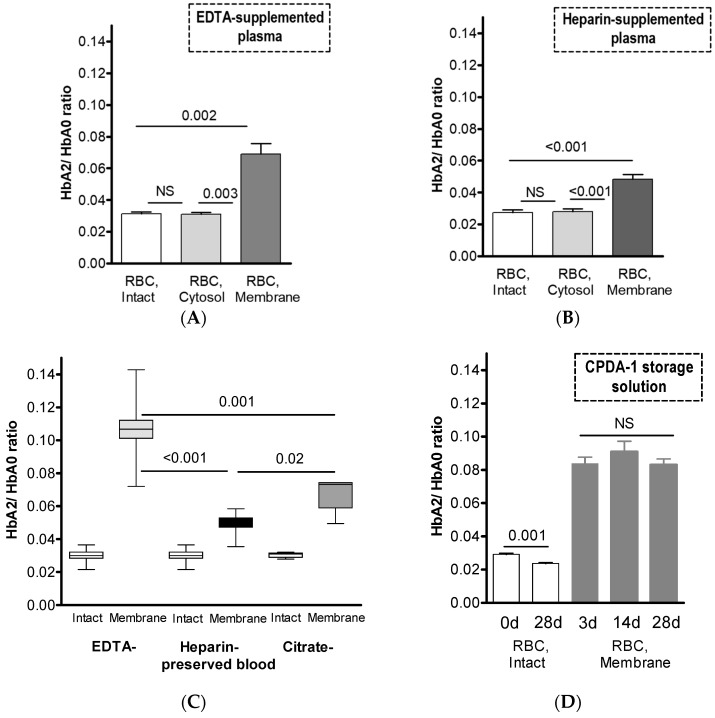
Effects of routinely used anticoagulants and CPDA-1 storage solution on Hb isoform distribution in RBCs. HbA2/HbA0 ratios in intact RBCs, and RBC cytosol and membrane of samples collected into (**A**) EDTA-supplemented (n = 6) or (**B**) heparin-supplemented (n = 8) tubes are shown. (**C**) HbA2/HbA0 membrane pools in RBCs collected in K_3_EDTA (n = 21), heparin (n = 16), and citrate (n = 5) tubes from different individuals. The blood samples were kept at room temperature prior to the tests. The total time period between the blood collection and the measurement did not exceed four hours. RBCs were isolated from plasma and buffy coat via short 1700× *g* centrifugation. Immediately after that, the measurement of Hb isoforms in intact RBCs and the isolation of RBC membranes (as described above) were performed. Wilcoxon signed-rank test was used to test significance for HbA2/HbA0 pools in the membranes of RBCs preserved in various anticoagulants; data are presented as median ± CI. Note the minimal differences (non-significant, NS) between intact RBCs’ HbA2/HbA0 ratios. (**D**) CPDA-1 study (n = 6), where cells were incubated at 4 °C for increasing periods to a maximum of 28 days, and HbA2/HbA0 ratios for intact and membrane RBC fractions were evaluated. The paired-matched data presented in panels (**A**,**B**,**D**) (means ± SD) were found to be normally distributed and were compared using paired Student’s *t*-test. Distributions of Hb isoforms (HbF, HbA0, and HbA2) corresponding to the data at current and next Figures are provided in the Appendix A.

**Figure 2 cells-12-02280-f002:**
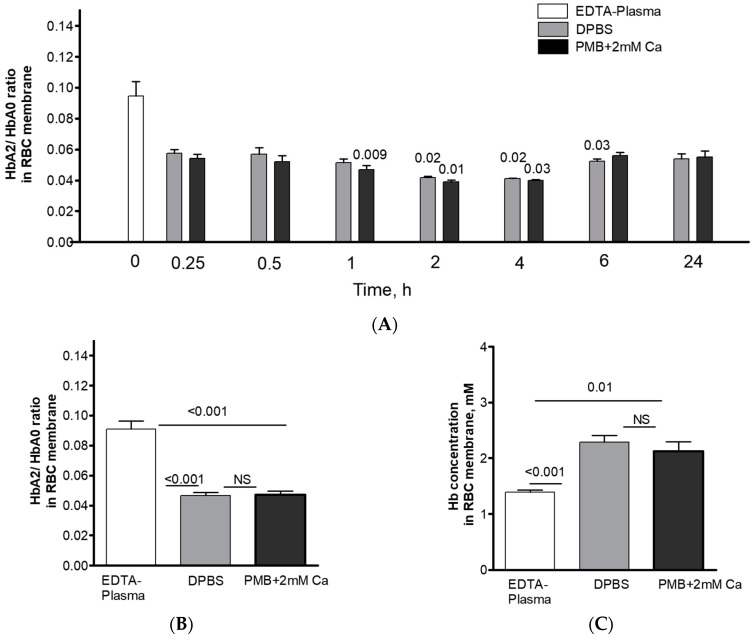
Changes in Hb concentration and isoform distribution in the membranes of RBCs incubated in cell-maintenance solutions DPBS and PMB. For time–response studies (**A**), RBCs from the same individuals (n = 3) collected in EDTA tubes were exposed to the media for 24 h with sampling at different time points. The HbA2/HbA0 ratio in intact RBCs was 0.032 ± 0.001. The matched comparison for (**B**) HbA2/HbA0 in membrane pools (n = 6) and (**C**) Hb concentration (n = 6) in intact RBCs and RBC membranes exposed to EDTA plasma, DPBS, and PMB (2 h, 37 °C) are shown. Results are presented as means ± SD. Significance for each presented set was determined using paired Student’s *t*-test at *p* ≤ 0.05; NS, not significant. Mean HbA2/HbA0 ratio in intact RBCs was 0.028 ± 0.003, while bulk Hb concentration (±SD) in intact RBCs made up 19.4 ± 0.76 mM.

**Figure 3 cells-12-02280-f003:**
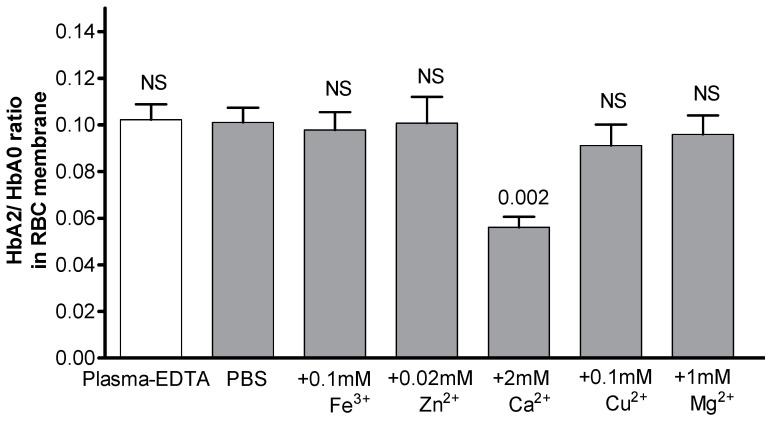
Ca^2+^, but not other bi-/trivalent cations, modifies membrane Hb isoform distribution. Cells from the same individuals were exposed to bi-/trivalent cation-free DPBS (i.e., commercially produced buffer with negligible contents of these electrolytes) supplemented with either Zn^2+^, Ca^2+^, Mg^2+^, Cu^2+^, or Fe^3+^ at near-physiological concentrations for 2 h at 37 °C prior to membrane isolation. Data are presented as means ± SD. Significance was determined for each presented set using paired Student’s *t*-test at *p* ≤ 0.05; NS, not significant. Mean HbA2/HbA0 ratio (±SD) in intact RBC was 0.030 ± 0.002.

**Figure 4 cells-12-02280-f004:**
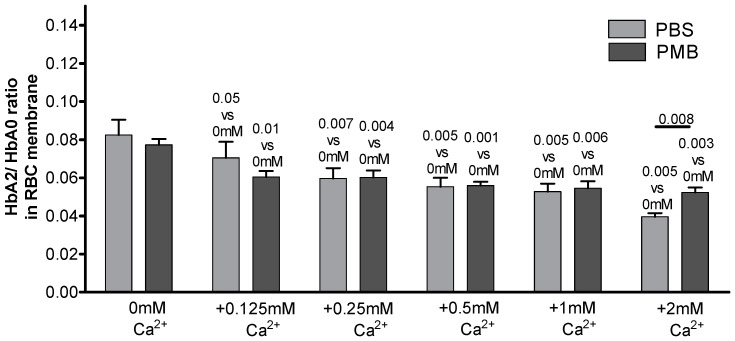
Dose response to extracellular Ca^2+^ of Hb isoform ratio in RBC membrane. Erythrocytes were exposed to increasing concentrations of Ca^2+^ pre-added to Ca^2+^-free DPBS (n = 6) or PMB (n = 4). Data are presented as means ± SD. Significance compared to the corresponding ‘0 mM Ca^2+^’ set of DPBS or PMB was determined using paired Student’s *t*-test at *p* ≤ 0.05; NS, not significant. Non-paired Student’s *t*-test was used to determine the significance of HbA2/HbA0 membrane pools in DPBS- vs. PMB-exposed RBCs for the same Ca^2+^ concentrations. Except for the significance shown for the ‘2 mM Ca^2+^’ sets (*p* = 0.008), no substantial differences for supplemented Ca^2+^ concentrations were noted for independent measurements of maintenance in DPBS vs. PMB. Mean HbA2/HbA0 ratios (±SD) in intact RBCs exposed to DPBS and PMB were 0.029 ± 0.003 and 0.032 ± 0.003, respectively.

**Figure 5 cells-12-02280-f005:**
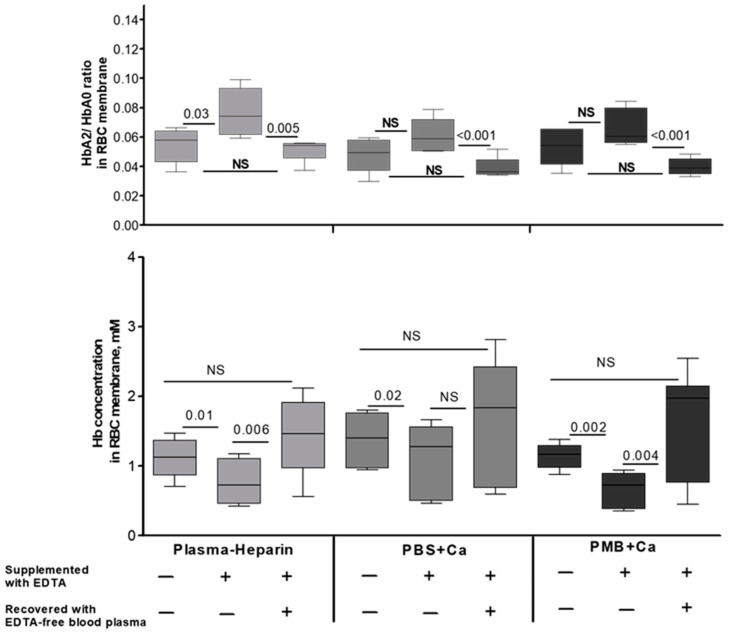
Involvement of additional extracellular factors in changes in membrane Hb concentration and isoform distribution under normo- and hypo-calcemic conditions. The erythrocytes were incubated in heparin-preserved plasma or 2 mM Ca^2+^-supplemented PBS or PMB (indicated by various colors) with or without 5 mM EDTA for 2 h at 37 °C. Erythrocytes were then quickly washed and incubated for an additional 2 h with EDTA-free plasma or buffers. The corresponding comparisons for the membrane-bound HbA2 pool (**upper** panel) and Hb concentration (**bottom** panel) are shown. Wilcoxon signed-rank test was used for statistical analysis; the data are presented as median ± CI. NS, not significant. Median HbA2/HbA0 ratio and Hb concentration in intact RBCs were 0.028 ± 0.003 and 20.7 ± 1.06 mM, respectively.

**Figure 6 cells-12-02280-f006:**
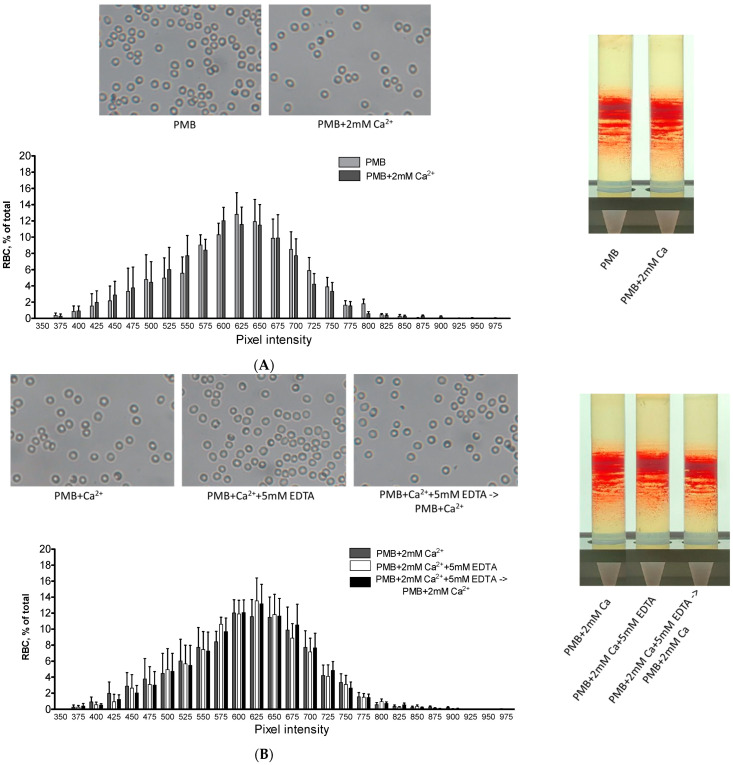
Extracellular Ca^2+^ has a minimal influence on RBC morphology and heterogeneity. The erythrocytes were preincubated in PMB with or without 2 mM Ca^2+^ or in Ca^2+^-supplemented PMB with or without 5 mM EDTA for 2 h at 37 °C. In the subsequent experiment, RBCs exposed to Ca^2+^-supplemented PMB with 5 mM EDTA were quickly washed with Ca^2+^-PMB and then incubated for another 2 h at 37 °C. In panels (**A**,**B**), pixeled projected areas were evaluated as described in Section 2.5. The datasets for 5 samples are presented as means ± SD, and no significant differences between the experimental groups were found. In corresponding studies, only a minimal effect of the treatments was observed on RBC heterogeneity (tested by RBC separation on a Percoll density gradient). Representative images for each experimental set are shown.

**Figure 7 cells-12-02280-f007:**
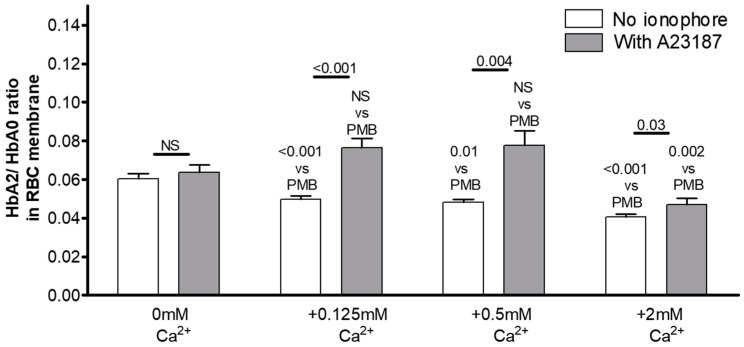
Effect of increasing concentrations of Ca^2+^ on Hb isoform distribution in the membranes of RBCs in the presence of Ca^2+^ ionophore A23187 (final 10 µM). Data are presented as average ± SD. Significance of values for RBCs exposed to increasing Ca^2+^ levels vs. those for ‘Ca^2+^-free PMB’ was evaluated. In addition, data for Ca^2+^-exposed RBCs in the presence vs. absence of the ionophore were compared.

**Figure 8 cells-12-02280-f008:**
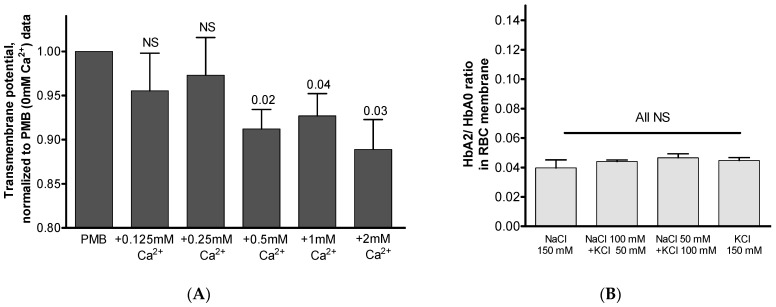
Possible influence of transmembrane potential. (**A**) Effect of increasing Ca^2+^ concentration on transmembrane potential was determined using means of the voltage-sensitive dye DiBAC4(3) and a flow cytometry approach as detailed in Section 2.7. Data were normalized to the ‘0 mM Ca^2+^’ value, and significance relative to this value is shown. (**B**) Membrane HbA2/HbA0 ratios in RBCs exposed to increasing fractions of KCl (0, 50, 100, and 150 mM) replacing the equivalent fractions of NaCl, in the presence of the K^+^ ionophore valinomycin (final 1 µM). Data are presented as means ± SD; minimal differences (all NS, *p* > 0.05) between all matching tests were examined using paired Student’s *t*-test. Mean HbA2/HbA0 ratio (±SD) in intact RBCs in these experiments was 0.031 ± 0.001.

**Figure 9 cells-12-02280-f009:**
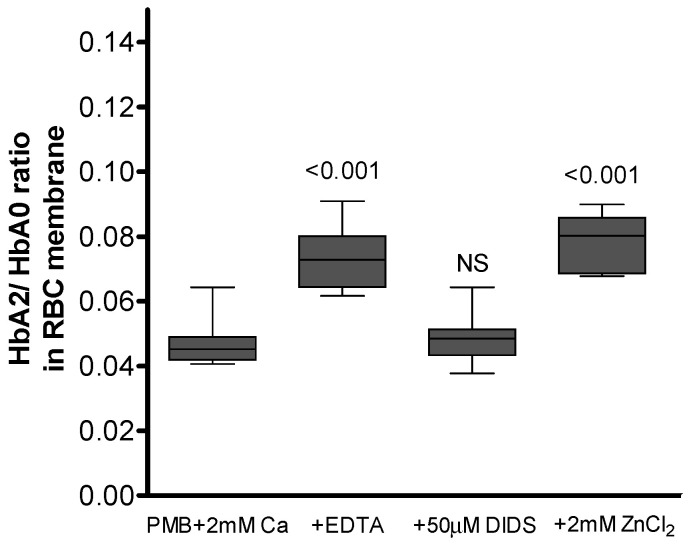
Possible involvement of AE-1 in the observed mechanism. Cells collected in heparin tubes were washed of plasma and treated in 2 mM of Ca^2+^-PMB with either 5 mM of EDTA, 50 µM of DIDS or 2 mM of ZnCl_2_. Wilcoxon signed-rank test was used for statistical analysis; data for 8 subjects are presented as median ± CI. Significance vs. control ‘PMB + 2 mM Ca^2+^’ dataset is presented, where NS relates to *p* > 0.05. Median HbA2/HbA0 ratio (±CI) in intact RBCs was 0.029 ± 0.001.

**Figure 10 cells-12-02280-f010:**
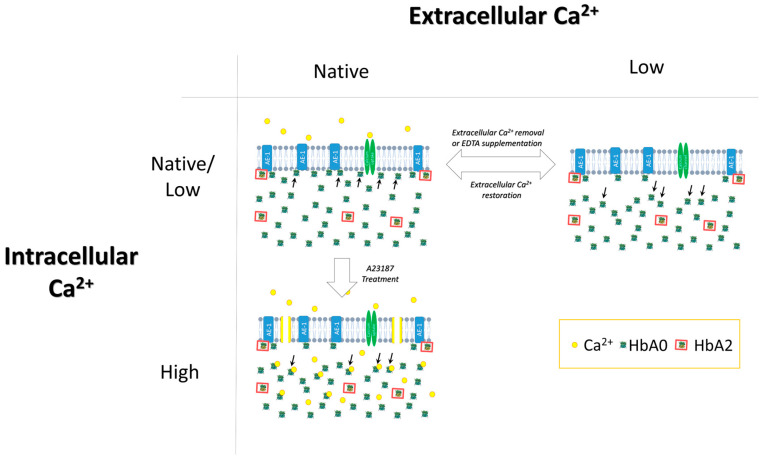
The proposed model.

**Table 1 cells-12-02280-t001:** Distribution of Hb isoforms (HbF, HbA0, and HbA2) in intact RBCs and in their membrane and cytosolic compartments. Blood samples collected into K_3_EDTA- or heparin-supplemented tubes (n = 6 and 8, respectively) were kept at room temperature prior to the experimental manipulations. The total time period between the blood collection and the measurement did not exceed four hours. RBCs were isolated from plasma and buffy coat via short centrifugation at 1700× *g*. Immediately after that, the measurement of Hb isoforms in intact RBCs was performed. Then, RBCs were lysed with ice-cold HEPES-based hypoosmotic solution, and hemolysates were electrophoresed to evaluate fractions of each Hb isoform in the RBC cytosol; the procedure was repeated three more times to obtain membranes for determination of Hb isoform distribution. Percent of each Hb isoform out of total Hb and HbA2/HbA0 ratios are shown. Data are presented as means ± SD. Significance (presented as superscript values) was determined compared to corresponding RBC membrane datasets using paired Student’s *t*-test at *p* ≤ 0.05; NS, non-significant.

	EDTA-Supplemented Plasma (n = 6)			Heparin-Supplemented Plasma (n = 8)		
Hb Isoforms		RBC			RBC	
	Intact	Membrane	Cytosol	Intact	Membrane	Cytosol
HbF, % of total Hb	0.30 ± 0.09 ^0.04^	0.17 ± 0.14	0.27 ± 0.08 ^NS^	0.36 ± 0.13 ^NS^	0.38 ± 0.16	0.36 ± 0.12 ^NS^
HbA2, % of total Hb	3.02 ± 0.28 ^0.002^	6.42 ± 1.46	2.98 ± 0.26 ^0.003^	2.65 ± 0.52 ^<0.001^	4.41 ± 0.72	2.60 ± 0.44 ^<0.001^
HbA0, % of total Hb	96.7 ± 0.26 ^0.001^	93.4 ± 1.33	96.8 ± 0.23 ^0.002^	97.0 ± 0.49 ^<0.001^	95.2 ± 0.67	97.0 ± 0.40 ^<0.001^
HbA2/HbA0	0.031 ± 0.003 ^0.002^	0.069 ± 0.017	0.031 ± 0.003 ^0.003^	0.027 ± 0.005 ^<0.001^	0.046 ± 0.008	0.027 ± 0.005 ^<0.001^

**Table 2 cells-12-02280-t002:** Chemical composition of the examined solutions.

	Na^+^/K^+^/Cl^−^, mM	Phosphates, mM	Mg^2+^/Ca^2+^/Zn^2+^, mM	Glucose, mM	Amino Acids	HEPES, mM	Trisodium Citrate, mM	Citrate, mM	Adenine, mM	Albumin, %	pH
Plasma (EDTA)	Native	Native	Minor (chelated)	Native	Native	–	–	–	Native	Native	7–7.4
Plasma (Citrate)	Native	Native	Minor (chelated)	Native	Native	–	–	–	Native	Native	7–7.4
Plasma (Heparin)	Native	Native	Native	Native	Native	–	–	–	Native	Native	7–7.4
CPDA-1	16/0/0	16	0	161	–	–	89.4	15.5	2	–	5.5
DPBS	146/4.1/141	1.5	0.5/0.9/0	–	–	–	–	–	–	–	7–7.4
PBS	146/4.1/141	1.5	0	–	–	–	–	–	–	–	7–7.4
PMB	140/4/144	–	0.75/2/0.015	10	Native	20	–	–	–	0.1	7–7.4

**Table 3 cells-12-02280-t003:** Effect of content and activity of plasma proteins on HbA2/HbA0 ratio. RBCs were incubated in autologous plasma preheated at 56 °C for 30 min or in PMB supplemented with 5% BSA for 2 h at 37 °C. Data are presented as means ± SD. Significance was determined using paired Student’s *t*-test at *p* ≤ 0.05; NS, non-significant.

	HbA2/HbA0 Ratio		
	Number	Intact	Membrane
EDTA-plasma	4	0.031 ± 0.003	0.101 ± 0.017
EDTA-plasma, heated at 56 °C			0.099 ± 0.014 ^NS^
EDTA-plasma	7	0.023 ± 0.002	0.072 ± 0.012 ^<0.001^
PMB			0.040 ± 0.006
PMB, supplemented with 5% BSA			0.040 ± 0.008 ^NS^

**Table 4 cells-12-02280-t004:** Effect of hypocalcemia on RBC membrane permeability and metabolic properties. Erythrocytes were preincubated in PMB with or without 2 mM Ca^2+^ or in Ca^2+^-supplemented PMB with or without 5 mM EDTA for 2 h at 37 °C. In the subsequent experiment, RBCs exposed to Ca^2+^-supplemented PMB with 5 mM EDTA were quickly washed with Ca^2+^-supplemented PMB and then incubated for another 2 h at 37 °C. All samples were then washed and supplemented with fresh buffer. Medium K^+^, glucose, and lactate contents were immediately determined via GEM^®^ Premier™ 5000 blood gas analyzer (0 h). Samples were then incubated for 4 h with gentle shaking at 37 °C and remeasured. The 4 h vs. 0 h difference was normalized to the total Hb concentration of the sample (tHb), which was measured at each time point. Significance was determined using paired Student’s *t*-test at *p* ≤ 0.05; NS, non-significant.

	Number	K^+^ Release, mM Normalized to tHb	Glucose Consumption, mM Normalized to tHb	Lactate Efflux, mM Normalized to tHb
PMB + 2 mM Ca^2+^ PMB	10	0.061 ± 0.017 0.082 ± 0.035 ^0.03^	1.866 ± 0.796 1.814 ± 0.649 ^NS^	0.184 ± 0.049 0.185 ± 0.038 ^NS^
PMB + 2 mM Ca^2+^	8	0.098 ± 0.036	1.614 ± 0.517	0.171 ± 0.021
PMB + 2 mM Ca^2+^ + 5 mM EDTA		0.188 ± 0.036 ^<0.001^	0.745 ± 0.806 ^0.02^	0.098 ± 0.032 ^<0.001^
PMB + 2 mM Ca^2+^ + 5 mM EDTA ->				
PMB + 2 mM Ca^2+^		0.099 ± 0.017	1.410 ± 0.657	0.137 ± 0.019 ^0.01^

**Table 5 cells-12-02280-t005:** Effect of increasing Ca^2+^ concentration in the external milieu on intracellular Ca^2+^ content in the presence or absence of Ca^2+^ ionophore A23187. Measurements were performed with the intracellular Ca^2+^ dye, Fluo-4 AM, and a flow cytometry approach. Results (normalized to ‘0 mM Ca^2+^’ test value) are presented as means ± SD. Significance for each test was determined vs. ‘0 mM Ca^2+^’ using paired Student’s *t*-test at *p* ≤ 0.05; NS, non-significant.

	Number	Fluo-4, Normalized to PMB Result
PMB	5	1
PMB + 0.125 mM Ca^2+^		1.136 ± 0.085 ^0.02^
PMB + 0.5 mM Ca^2+^		1.178 ± 0.091 ^0.01^
PMB + 2 mM Ca^2+^		1.174 ± 0.098 ^0.02^
PMB + 10 µM A23187		0.946 ± 0.088 ^NS^
PMB + 10 µM A23187 + 0.125 mM Ca^2+^		0.878 ± 0.105 ^NS^
PMB + 10 µM A23187 + 0.5 mM Ca^2+^		5.223 ± 0.427 ^<0.001^
PMB + 10 µM A23187 + 2 mM Ca^2+^		6.63 ± 0.731 ^<0.001^

## Data Availability

Data are contained within the article. The data presented in this study are available upon request from the corresponding author.

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
