# Peer review of "The Impact of Ca2+ on Intracellular Distribution of Hemoglobin in Human Erythrocytes"

_cells, 2023, doi:10.3390/cells12182280_

Round 1

Reviewer 1 Report

The reviewed manuscript is devoted to an interesting and important topic corresponding to the issue of the journal. New important results were obtained. At the same time it is possible to make some comments and notes to the manuscript. They are written in the attached file. 

Most of these notes ate technical and can be corrected rather easily. After corrections the paper can be published in the journal.

Reviewer 2 Report

Livshits et al. examine hemoglobin (Hb) binding to the red blood cell (RBC) membrane in human blood samples drawn collected under various anti-coagulants. They find that extracellular Ca2+ depletion by chelators or use of Ca2+ free extracellular membrane leads to membrane-bound Hb release and redistribution of HbA0 and HbA2 between membrane and cytosol, favoring HbA2 bound to the membrane. Reintroduction of Ca2+ into the extracellular medium reverses the redistribution of membrane bound HbA0 compared with HbA2. Docking sites in the cytosolic domain of anion exchanger 1 (AE-1) protein are suggested to regulate this Ca2+ dependent membrane bound Hb distribution. These data illustrate the influence of specific blood sample anticoagulants on RBCs that may have clinical significance.

Comments:

Hemoglobin oxygen saturation of stored blood exhibits individual variation such as male vs female donors and oxygen saturation can influence membrane-bound Hb fraction. What was the variation in hemoglobin oxygen saturation of the blood samples at time of measurement? Does hemoglobin oxygen saturation affect the HbA2/HbA0 ratio? 

Table 1 legend indicates that Hb isoforms in intact RBCs were measured immediately after RBC isolation.  Does this mean immediately after blood draw or after plasma and leucocytes were removed?  What was the time duration from blood draw to the beginning of the measurements and at what temperature were samples stored until beginning of the measurements? 

What proportion of the total hemoglobin is bound to membrane compared with amount in the cytosol in the current study? Is this similar for EDTA and heparin collected samples?

Figure 1a,b,c: What temperature was used for the data provided in Figure 1? Were all samples incubated at 4°C prior to determinations of HbA2/HbA0 ratio? If so, what was the incubation time for (a), (b) and (c)? Does prior incubation at 4°C, 30°C or 37°C affect the HbA2/HbA0 ratio?

Figure 1d: Do the authors think that temperature can influence the change % HbA2 in intact RBCS with time in storage? Decrease in % HbA2 in stored blood samples has been reported in a Letter to the Editor and should be cited and discussed further (Hildrum JM, Fjeld B, Risahagen SM, Bernatek BJ, Klingenberg O. Assessment of Hemoglobin A2 stability at room temperature during 24 or 25 days as measured by high pressure liquid chromatography and capillary electrophoresis. Int J Lab Hematol. 2021;43:e266–e270.https://doi.org/10.1111/ijlh.13555). 

Figure 2: How does temperature affect the HbA2/HbA0 ratio provided in Figure 2? What was the temperature used for data provided in (a)?

Please clarify the statement: " Intriguingly, the observed alteration in the pre-membrane HbA2 fraction was associated with a significant rise in the concentration of total Hb in the pre-membrane Hb pool in both DPBS and PMB (Figure 2C). This is not the MCHC (mean corpuscular hemoglobin concentration) that should be about 34 g/dl. How is the Hb concentration in the RBC membrane determined? 

Please clarify: " Hb pool with HbA2 cannot be explained by the variation in extracellular concentrations of bi-/trivalent cations, and specifically Ca2+ and Mg2+" since the authors then provide data for a role for Ca2+.

Please clarify the statement: "The effect was much less pronounced with 2 mM Ca2+ in the medium, but could be explained by the time-dependent instability of A23187 in presence of elevated extracellular Ca2+."

Figure 6: How does the Percoll density gradient change between 30°C and 37°C?

What temperature was used for incubation in the labeling experiments?

Reviewer 3 Report

Minor editing of English language required.

Round 2

Reviewer 2 Report

Livshits et al. show the influence of extracellular Ca2+ on red blood cell membrane associated HbA2/HbA0 ratio with minimal impact on cell morphology or heterogeneity. Measurements include change in the HbA2/HbA0 ratio with time for cells drawn in EDTA tubes and then incubated in PBS with Ca2+ and Mg2+ or plasma-mimicking buffer. These observations may impact on the selection of blood sample anticoagulant.

The authors have addressed a number of concerns raised in the original review with additional clarifications to the text including methodology and discussion, and additional Supplementary data. Supplementary data include estimates of the proportion of Hb content in RBC membrane vs intact Hb fraction and temperature independence of th HbA2/HbA0 ratio in the pre-membrane pool.

Reviewer 3 Report

 Accept in present form